# Ranking the biases: The choice of OTUs vs. ASVs in 16S rRNA amplicon data analysis has stronger effects on diversity measures than rarefaction and OTU identity threshold

Marlène Chiarello[1]*, Mark McCauley[1], Sébastien Villéger[2], Colin R. Jackson[1]

**1** Department of Biology, University of Mississippi, University, MS, United States of America, **2** MARBEC, University of Montpellier, CNRS, Ifremer, IRD, Montpellier, France

* marlene.chiarello@gmail.com

**Data Availability Statement:** Bacterial sequences for this paper are available in the NCBI Sequence Read Archive database (http://www.ncbi.nlm.nih.

## Abstract

Advances in the analysis of amplicon sequence datasets have introduced a methodological shift in how research teams investigate microbial biodiversity, away from sequence identity-based clustering (producing Operational Taxonomic Units, OTUs) to denoising methods (producing amplicon sequence variants, ASVs). While denoising methods have several inherent properties that make them desirable compared to clustering-based methods, questions remain as to the influence that these pipelines have on the ecological patterns being assessed, especially when compared to other methodological choices made when processing data (e.g. rarefaction) and computing diversity indices. We compared the respective influences of two widely used methods, namely DADA2 (a denoising method) vs. Mothur (a clustering method) on 16S rRNA gene amplicon datasets (hypervariable region v4), and compared such effects to the rarefaction of the community table and OTU identity threshold (97% vs. 99%) on the ecological signals detected. We used a dataset comprising freshwater invertebrate (three Unionidae species) gut and environmental (sediment, seston) communities sampled in six rivers in the southeastern USA. We ranked the respective effects of each methodological choice on alpha and beta diversity, and taxonomic composition. The choice of the pipeline significantly influenced alpha and beta diversities and changed the ecological signal detected, especially on presence/absence indices such as the richness index and unweighted Unifrac. Interestingly, the discrepancy between OTU and ASV-based diversity metrics could be attenuated by the use of rarefaction. The identification of major classes and genera also revealed significant discrepancies across pipelines. Compared to the pipeline's effect, OTU threshold and rarefaction had a minimal impact on all measurements.

## Introduction

Evaluating the microbial diversity of various environments, from host-associated microbiomes to free-living communities in water, soil, and air, is essential for understanding biodiversity and improving human health and agriculture [1, 2]. The development and increased

gov/bioproject/740316), under the BioSample numbers SAMN19838587-SAMN19838803.

**Funding:** Funding was provided from National Science Foundation (https://www.nsf.gov/) award DEB 1831531 to C.R.J. The funders had no role in study design, data collection and analysis, decision to publish, or preparation of the manuscript.

**Competing interests:** The authors have declared that no competing interests exist.

accessibility of high-throughput sequencing technologies [3], has supported the advancement of large-scale assessments of microbial diversity over the past decade [4]. The most popular sequencing technique for analysis of bacterial diversity is sequencing a specific gene (or region of a gene, e.g. a hypervariable region of the bacterial 16S rRNA gene) using Polymerase Chain Reaction to create sequences called amplicons (targeted amplicon sequencing), which can be done using diverse platforms (e.g. 454, Illumina, IonTorrent). One of the most popular is Illumina MiSeq, producing paired reads, *i.e.* sequencing targeted loci from both its 5' (forward) and 3' (reverse) extremities. As with improvements in sequencing technology, bioinformatics techniques for the analysis of high-throughput sequencing data have been concurrently improving [5]; for targeted amplicon sequencing alone, there are now at least 15 dedicated pipelines available [6–11].

In order to investigate biodiversity amplicon sequencing data, researchers usually attempt to aggregate the >50,000 paired reads into sequences (*e.g.* Illumina MiSeq delivers up to 15Gb using kits for 2x300b reads), before grouping them into clusters, widely termed operational taxonomic units (OTUs). OTUs are defined as a cluster of sequences that have a sequence identity above a given threshold, often set to 97% [6]. Clustering based on 97% identity reduces the size of the raw 16S rRNA dataset, and decreasing the computational requirements for analysis [12]. It also reduces the impact of sequencing errors in downstream diversity estimations, as erroneous sequences are likely to be merged with correct sequences [10, 13].

Recently, a methodological shift has occurred with the increased use of denoising methods, producing exact sequence variants or amplicon sequence variants (ASVs, [14]) instead of producing clusters. Denoising methods generate an error model based on the quality of the sequencing run and use this model to distinguish between the predicted "true" biological variation and that likely generated by sequencing error [7, 15]. Remaining 'true' sequences varying from as little as one single nucleotide are then defined as separate ASVs, which is intended to allow for a higher accuracy than 97% identity OTUs. However, given their construction method, ASVs are not equivalent to "100%-OTUs" [16].

With the increasing popularity of denoising approaches (hereafter, 'ASV-based methods'), studies have compared the results from clustering ('OTU-based') and ASV-based approaches. Based on mock communities, ASV-based approaches had a higher sensitivity in detecting the bacterial strains present, sometimes at the expense of specificity [17–20]. However, studies utilizing soil, rhizosphere, and human microbiome datasets found similar overall biological signals [14, 21, 22]. From these studies, the main weakness of OTU-based approaches appears to be in the accurate detection of alpha diversity, as OTUs often overestimate bacterial richness when compared to ASVs, whereas estimates of beta diversity estimates are typically more congruent between methodologies [15, 19, 21, 22]. Further, an analysis using various OTU and ASV-based programs and a wide range of natural and mock communities identified significant discrepancies in the taxonomic assignment and estimation of relative abundance of functionally important taxa [23].

Despite these studies, the drivers of discrepancies between OTU and ASV approaches have not been fully disentangled. For instance, what is the influence of choosing ASVs- vs. OTUs-based approaches when compared to the influence of other methodological choices that are known to impact biological patterns, such as rarefaction, OTU identity threshold, and specific diversity indices [24–27]? Further, while the relative impact of each of these choices may depend on the characteristics (richness, evenness of relative abundances, phylogenetic structure) of the community analyzed, many of the aforementioned studies focused on only one environment (but see [17, 23]). A bacterial community with a few closely related dominant taxa may be more sensitive to the choice of sequence processing method than a community with greater phylogenetic diversity or evenness of abundances. Lastly, while phylogenetic

diversity indices (e.g. UniFrac) are typically more efficient at detecting ecological patterns and evolutionary processes than indices based on just operational units (e.g. Bray-Curtis or Jaccard indices, hereafter termed 'taxonomic diversity indices') [26, 28, 29], only one of the aforementioned studies compared phylogenetic diversity metrics obtained from ASVs vs. OTUs datasets [23].

In this study, we compare the influence of two widely used sequence-processing methods, namely the ASV-based DADA2 and the OTU-based MOTHUR pipeline, on the diversity, ecological and compositional patterns of the bacterial communities present in three different microbial assemblages (aquatic sediment, particle-associated freshwater communities 'seston', and the freshwater mussel gut microbiome). We compared the effects of ASV vs. OTU approaches using two different identity thresholds to those with varying levels of sequence rarefaction, and eight contrasting alpha and beta-diversity metrics.

## Methods

### Sampling of bacterial communities

A total of 54 surface sediment, 54 seston, and 119 mussel (host-associated) microbiome samples were collected from 18 sample sites located on six rivers in the Tennessee and Mobile River Basins, USA, in Summer 2019 (S1 Table in S2 File). At each site, three sediment and three seston samples were collected in the middle of the river channel. Sediment samples were collected using sterile 15 mL centrifuge tubes that were dragged through the top 5 cm of sediment. Seston samples were collected by filtering 120 mL of river water through sterile 47 mm, 1 μ pore size sterile glass fiber filters, which were then placed in sterile 15 ml tubes. 4–5 mussel specimens belonging to three different species (*Lampsilis ornata*, *Amblema plicata*, *Cyclonaias asperata*) were manually collected (S1 Table in S2 File). Mussels were collected under the authority of the permit (USFWS permit #TE68616B-1 and ALCDNR permit #2016077745468680) issued by the Alabama Department of Conservation and Natural Resources to Dr. Carla Atkinson, University of Alabama. After collection, all filters, sediment samples, and mussels were stored on ice and taken to the University of Alabama for storage at -80˚C. Mussel gut tissue was excised using sterile dissecting equipment, and all samples were transported to the University of Mississippi for microbiome analysis.

### DNA extraction and 16S rRNA gene amplicon sequencing

DNA was extracted from mussel gut tissue using a PowerSoil Pro extraction kit (Qiagen, Germantown, MD, USA) as described previously [30]. For seston and sediment microbiome analysis, filters or 30 mg of sediment were added directly into the bead beating tube of the extraction kit, before performing the same procedure.

Dual-indexed barcoded primers were used to amplify the V4 region of the 16S rRNA gene of the extracted DNA following established techniques [6, 30] using primers from [31]. The amplified 16S rRNA gene fragments were combined and spiked with 20% PhiX before being sequenced on an Illumina MiSeq at the University of Mississippi Medical Centre Molecular and Genomics Core facility. Sequence data were obtained as FASTQ files, which were further processed using Mothur and DADA2. Sequencing quality was assessed using fastqc reports and was summarized using 'fastqcr' R-package [32] and provided in S3 Table in S2 File.

### OTU-based pipeline (Mothur)

We used Mothur v1.8.0 [33] to assemble, read, and filter the obtained sequences, before clustering them into OTUs. We followed the recommended procedures for Illumina MiSeq

sequence data, as it is the methodology most likely to be used by researchers (https://mothur. org/wiki/miseq_sop/, December 2020). Briefly, after merging the forward and reverse reads, we screened the sequences and removed those of unusual length or that contained ambiguous bases. Unique sequences were then aligned to the Silva 16S rRNA gene database v. 138 [34], and poorly aligned reads were removed. Sequences were classified using the Wang method [35] in Mothur through the same database to remove all non-prokaryotic sequences and sequences that were not classified to the Kingdom level. Chimeras were determined and removed using the *chimera.vsearch* command with default parameters. OTUs were assembled based on 97%- and 99%-identity, and OTU tables were constructed before retrieving the consensus classification for each OTU. The most abundant sequence within each OTU was selected as the reference sequence for further phylogenetic analyses. A total of 31,453 97%-OTUs and 58,483 99%-OTUs were detected in the unrarefied dataset of 217 samples, based on 3,420–143,827 valid sequence reads per sample. According to the current SOP, no abundance-based filtering of OTUs was applied before performing rarefaction.

## ASV-based pipeline (DADA2)

We used DADA2 R-package v1.20 [7] on R v3.6.3 [36] following the general tutorial available on the Github of the software (https://benjjneb.github.io/dada2/tutorial.html, November 2020). Forward and reverse reads with $>2$ or $>5$ estimated errors, respectively, were filtered and were truncated at the 3' end, where read quality dropped below a quality score of 2 (TrunQ = 2). After separate estimations of error rates on forward and reverse reads, ASVs were predicted and merged using a minimal overlap of 12 bases. Chimeras were removed using the consensus method with the *removeBimeraDenovo*() function. ASVs with $<243$ or $>263$ base pairs were removed. ASVs were classified using Wang's classifier [35] and the SILVA 16S rRNA gene database v. 138 [34], using DADA2's default parameters. There were a total of 21,606 ASVs in the unrarefied dataset of 217 samples, based on 3,164–112,811 valid sequence reads per sample. Despite the same version of the SILVA database utilized for classification of both OTUs and ASVs, minor differences were observed, with Oxyphotobacteria considered as a class within the ASV dataset, and Planctomycetes in the OTU datasets named as Planctomycetacia in the ASV dataset. To simplify comparison across methodologies, we renamed Oxyphotobacteria as Cyanobacteria, and Planctomycetacia as Planctomycetes within the ASV dataset.

**Relative abundance normalization and taxonomic diversity.** ASV and OTU tables were imported into R v3.6.3 [36] using the 'phyloseq' R-package v1.30 [37]. The minimal number of sequences within samples before rarefaction were 3164, 3420, and 3400, downstream DADA2 (ASVs) and Mothur (97%- and 99%-OTUs), respectively (S2 Table in S2 File). 12% and 4% of samples have less than 4,000 sequences in ASV-based and OTU-based datasets, respectively. Therefore, the three levels of rarefaction chosen were 1000, 2000, and 3000 sequences in order to compare both DADA2 and Mothur methodologies without losing too many samples. Rarefaction was performed by random subsampling sequences using function "rarefy_even_depth" from 'phyloseq' R-package. Subsequent analyses were performed at each rarefaction level as well as on unrarefied 97%-OTUs, 99%-OTUs, and ASVs tables, for a total of 12 community tables (three ASV/OTU approaches x four levels of rarefaction). Sampling coverage was determined for all analysis tables using Chao's coverage index provided in the 'entropart' R-package v1.6–8 [38]. Indices of taxonomic alpha diversity (observed richness, iChao1 and ACE richness indices correcting for unsampled operational units, and Shannon alpha diversity) were assessed using 'vegan' R-package v2.5–7 (for observed richness and Shannon) [39], 'entropart' (for iChao1) and 'fossil' R-package v0.4 (for ACE) [40]. Following recommendations from Jost [41], all alpha diversity indices were expressed in an equivalent number of operational units (*i.*

*e*., equivalent numbers of species, also known as Hill numbers). The evenness of the abundance of operational units was measured using Bulla's *O* index, which is less affected by the number of species than other evenness indices [42, 43].

Taxonomic beta diversity was assessed using complementary indices based on Jaccard and Bray-Curtis dissimilarities, using 'vegan'. While Jaccard is solely based on the presence/absence of operational units, Bray-Curtis gives more weight to the most abundant operational units.

## Phylogenetic diversity

ASV and OTU reference sequences were incorporated into the GreenGenes 99% phylogenetic tree v13.8 [44], using SEPP software [45] implemented in QIIME2 [46] using default parameters. The obtained bacterial phylogenetic tree was then pruned using '*ape*' R-Package v5.5 [47] to remove all tree leaves absent from our dataset while maintaining the tree's structure. This method allowed for a more accurate tree structure than a de-novo tree reconstruction using short sequences. Phylogenetic beta diversity was then assessed using the weighted (W-) and unweighted (U-) versions of the Unifrac index, using 'GUniFrac' R-package v1.3 [48].

## Statistical analyses

Statistical analyses were computed using R, and data visualization was made using 'ggplot2' R-package v3.3.5 [49]. Differences in alpha and beta diversity values across different methodologies (ASVs vs. 97%-OTUs vs. 99%-OTUs), rarefaction level (no rarefaction vs. rarefaction to 1,000 vs. 2,000 vs. 3,000 sequences), and OTU threshold (99% vs. 97% identity) were assessed using pairwise Wilcoxon signed-rank tests allowing for the identification of a significant difference across methodologies. Spearman's and Pearson's correlations in alpha diversity metrics were computed to measure the correlation strength for these values across methodologies. Similarly, correlations between beta diversity indices were measured using Spearman's and Pearson's correlations in Mantel tests available in 'vegan'. The correlation of relative abundance of taxonomic groups across the different analysis treatments was made using Spearman's signed rank tests. Similarly, correlations of the abundances of top bacterial classes and genera were made using separate Spearman's signed rank tests and plotted using 'corrplot' R-package v0.90 [50].

Detection of ecological signals (effect of community type, river, and sample site) in alpha diversity was conducted with Kruskal-Wallis tests and associated post-hoc pairwise comparisons using 'pgirmess' R-package v1.7.0 [51].

Detection of the same ecological effects on the structure of the microbial communities was conducted using separated PERMANOVAs performed on beta-diversity matrices using the 'vegan' R-package (900 permutations). The intensity of the signal was measured using the R-squared ($R^2$) obtained in PERMANOVA's outcomes. Such values were compared across treatments to assess the respective impact of ASVs vs. OTUs, rarefaction, and the beta diversity metric used on PERMANOVA's result, using random forest models provided in 'randomForest' R-package v4.6–14 [52]. These models are based on successive decision trees to allow ranking of the effects of several correlated variables on a given response [53], here on the intensity of the detected ecological signal.

# Results

## Structure and coverage of 16S rRNA datasets

ASV and OTU-based analyses resulted in significantly different numbers of operational units in non-rarefied datasets, with fewer ASVs (mean±standard deviation 303±215 per sample) than 97%-OTUs (1,386±1,026), and 99%-OTUs (1,708±1,319) (Wilcoxon signed-rank tests,

p<0.001 in all comparisons), despite a better coverage of ASV-based data based on rarefaction curves (S1-S3 Figs in S1 File). Before rarefaction, Chao's coverage was significantly lower in OTU-based datasets (93.7±5.6% and 91.5±7.8% respectively in 97%- and 99%-OTUs) than in the ASV-based dataset, where it averaged 100±0% (Wilcoxon signed-rank tests, p<0.001 between OTUs and ASVs, P>0.05 between 97%- and 99%-OTUs). Accordingly, after rarefaction, the number of species in each sample was close to the saturation value on rarefaction curves for the ASV datasets (coverage of 98.0±1.8% after rarefaction to 2000 sequences), but was well below the saturation point based on 97%- and 99%-OTUs (Coverage of 85.8±10.3% and 83.6±12.6% after rarefaction to 2000 sequences) (S1-S3 Figs in S1 File).

Before rarefaction, the abundance distribution of operational units was more even in the ASV dataset than in the 99%-OTU dataset and 97%-OTU datasets (Fig 1; Wilcoxon signed-rank tests p<0.001 in all comparisons). This pattern was apparent across all sample types, although evenness was highest in the sediment, intermediate in seston, and lowest in host-associated communities (Fig 1; Kruskal-Wallis and post-hoc associated pairwise tests, p<0.001 for all comparisons).

## Alpha diversity metrics

All alpha diversity metrics were distinct between sequence processing methods (ASVs vs. OTUs), at every rarefaction level (Wilcoxon signed-rank tests p<0.001 in all comparisons, Fig 2, S3 Fig in S1 File). ASV-derived richness was roughly twice as low as 99%-OTU and 97%-OTU richness, which were closer to each other (99%-OTU richness was 1.1 times higher than 97%-OTU richness). The difference between ASV- and OTU-derived richness was even higher using iChao1 (4.8–5.3 higher in 97%- and 99%-OTUs datasets than in ASVs, respectively) and ACE (3.5–4.1 times respectively) (S3 Fig in S1 File). However, differences between the approaches were lower in the case of Shannon's alpha diversity index (1.4 and 1.7 times respectively, Fig 2).

The ranks of alpha diversity metrics were more congruent across ASV- vs. OTU-based approaches than the values themselves (Table 1, higher correlations based on Spearman's than on Pearson's correlations). Correlations between ASVs and OTUs approaches were the lowest for richness indices that correct for sampling bias (iChao1 and ACE; Spearman's r = 0.61–0.84 depending on the index, rarefaction level, and OTU threshold), intermediate for observed richness (0.80–0.85), and highest for Shannon diversity (0.90–0.97) (Table 1). Rarefaction increased the strength of these correlations (Table 1).

Alpha diversity was highest in the sediment, intermediate in the seston, and lowest in mussel host-associated communities, regardless of sequence processing method or rarefaction level (Kruskal-Wallis and associated pairwise post-hoc tests, p<0.05 in all cases and for all comparisons, Fig 2). However, the effects of river and site varied with the sequence processing method and the metric used, with an intensity that was dependent on the type of the community (Fig 3). While differences in Shannon diversity between rivers were consistent regardless of the sequence processing method and sample type, differences in observed richness across rivers were more variable. For sediment samples, richness differences were generally stable to shifting from ASV to OTU-based approaches (*i.e.* no significant differences between rivers). However, for seston and mussel host communities, the selection of an ASV or OTU-based method influenced the outcome as to whether any rivers were significantly different in richness or not (Fig 3).

## Beta diversity indices

Beta diversity metrics computed with ASVs and OTUs were highly correlated (Mantel test, Spearman's *r* >0.9), with the exception of U-Unifrac (<0.81, Table 2). Beta diversity metrics based on 99%-OTUs were more correlated than those from ASVs than 97%-OTUs (Table 2).

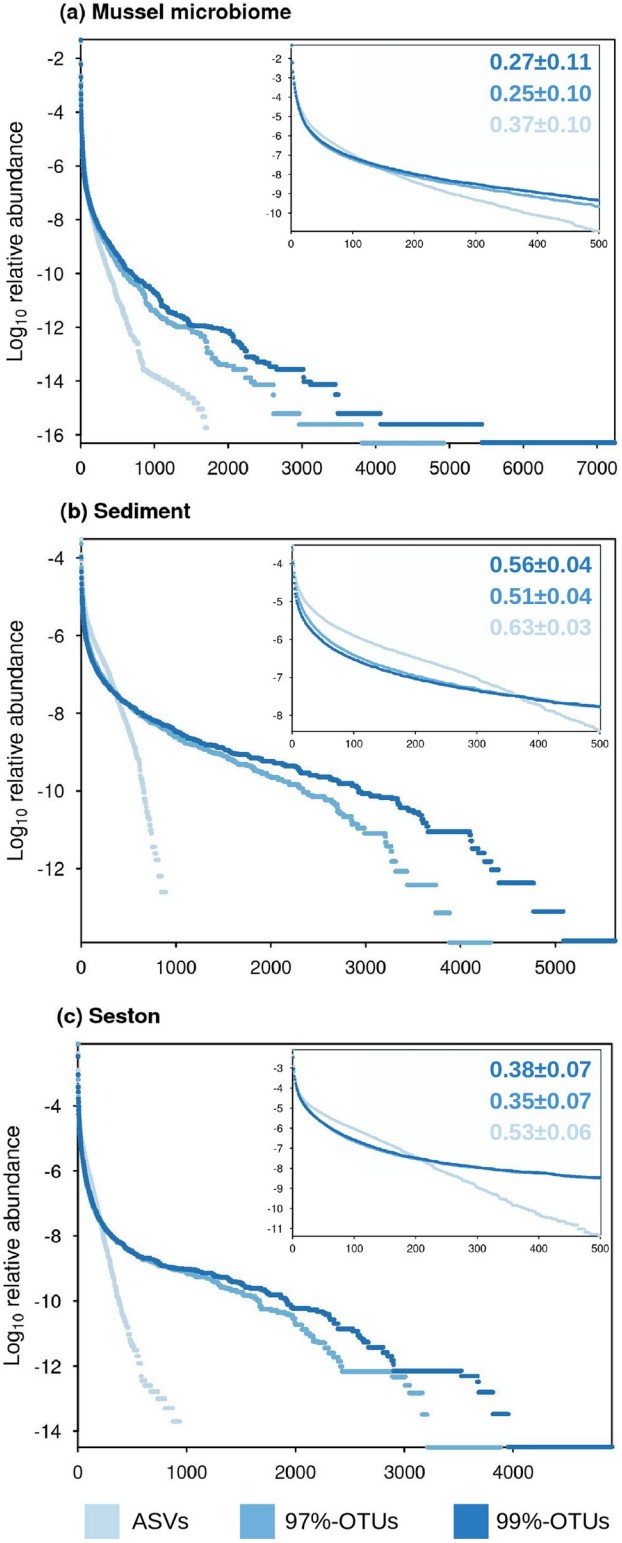

**Fig 1. Effect of sequence processing methodology on the rank abundance of operational units.** The relative abundance of ASVs (obtained from DADA2) and OTUs defined by 97% or 99% identity (obtained from MOTHUR). Units were sorted according to their rank of abundance and represented separately for each community type (a: mussel gut microbiome, b: sediment, c: seston). Inserts focus on the 500 most abundant operational units. The evenness of the relative abundance of the operational units computed Bulla's *O* are displayed on each plot.

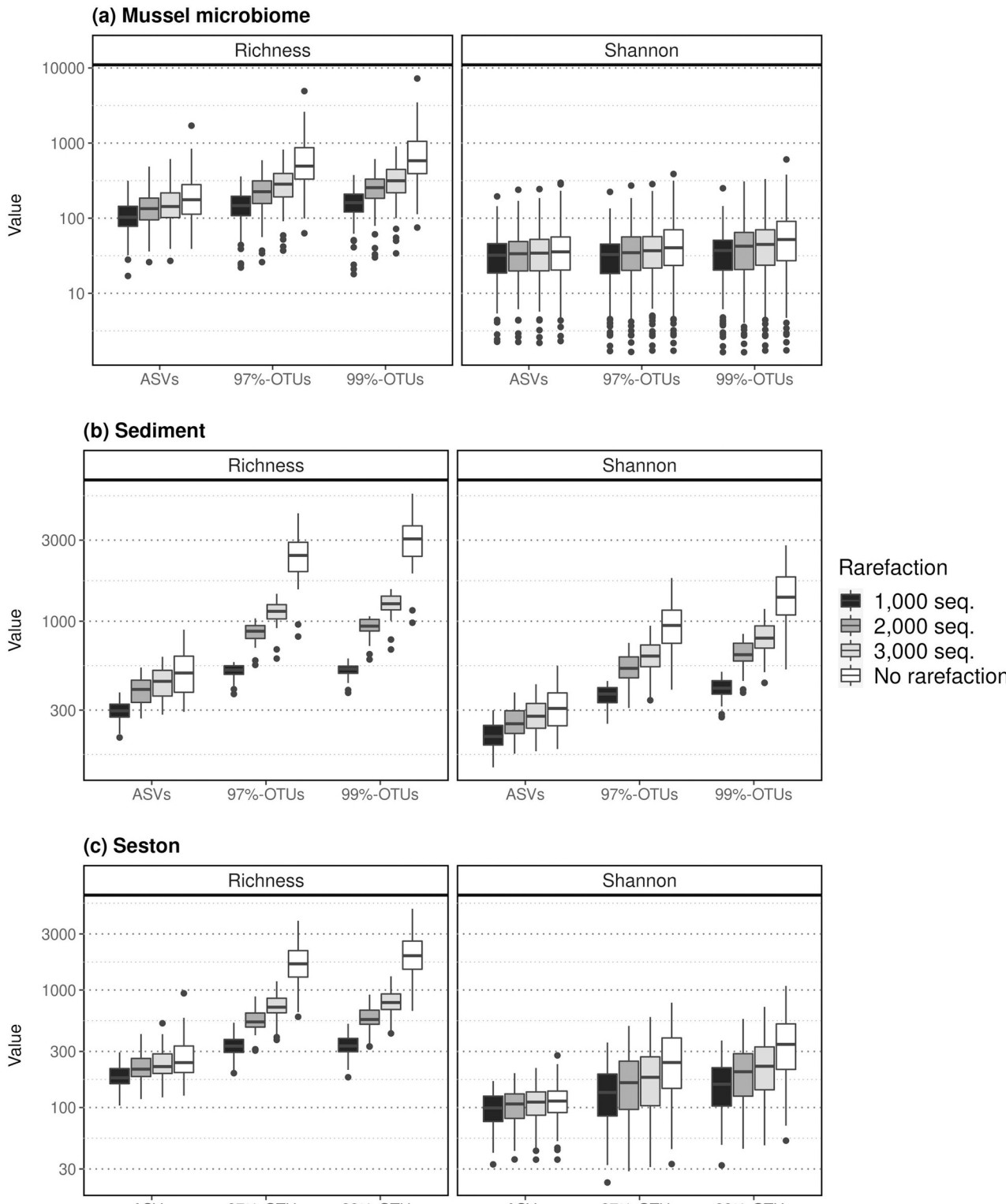

**Fig 2. Effect of sequence processing methodology on alpha diversity metrics for microbiome analyses.** Measure of taxonomic richness (expressed as the number of OTUs or ASVs and Shannon alpha diversity depending on the methodology used, namely ASV, 97%-OTU, and 99%-OTU, at three different levels of rarefaction (1,000; 2,000 and 3,000 sequences per sample) within each sample type studied (a-c). To ease the visualization of differences across methods and rarefaction levels, the y-axis of plots has been log transformed. The effect of methodological choices on other alpha diversity metrics is available in S3 Fig in S1 File.

**Table 1. Correlation between alpha diversity metrics between OTU- and ASV-based datasets across every rarefaction level tested.**

| Diversity index | Rarefaction | ASVs vs. 97%-OTUs | ASVs vs. 99%-OTUs |
|---|---|---|---|
| **Richness** | No rarefaction | **0.80 / <u>0.75</u>** | **0.81 / <u>0.77</u>** |
| | 3,000 sequences | **0.81 / <u>0.77</u>** | **0.81 / <u>0.78</u>** |
| | 2, 000 sequences | **0.85 / 0.82** | **0.85 / 0.83** |
| | 1,000 sequences | 0.90 / 0.90 | 0.91 / 0.90 |
| **Shannon** | No rarefaction | 0.95 / **0.87** | 0.96 / **0.87** |
| | 3,000 sequences | 0.96 / 0.91 | 0.96 / 0.91 |
| | 2, 000 sequences | 0.96 / 0.92 | 0.96 / 0.93 |
| | 1,000 sequences | 0.96 / 0.95 | 0.97 / 0.96 |
| **iChao1** | No rarefaction | **0.62 / <u>0.51</u>** | **0.61 / <u>0.49</u>** |
| | 3,000 sequences | **0.65 / <u>0.58</u>** | **0.66 / <u>0.58</u>** |
| | 2, 000 sequences | **0.70 / <u>0.64</u>** | **0.69 / <u>0.61</u>** |
| | 1,000 sequences | **0.76 / <u>0.72</u>** | **0.78 / <u>0.74</u>** |
| **ACE** | No rarefaction | **0.70 / <u>0.63</u>** | **0.69 / <u>0.61</u>** |
| | 3,000 sequences | **0.73 / <u>0.66</u>** | **0.73 / <u>0.64</u>** |
| | 2, 000 sequences | **0.77 / <u>0.72</u>** | **0.77 / <u>0.68</u>** |
| | 1,000 sequences | **0.84 / <u>0.77</u>** | **0.84 / <u>0.75</u>** |

Correlations were tested using Spearman's and Person's correlation tests (all P values were < 0.001). Correlations estimates are indicated as follows: Spearman's *r* / Pearson's *r*.

Correlations showing poorer agreement between OTUs and ASVs are noted, *i.e.* $0.8 < r < 0.9$ (**bold**) or $r < 0.8$ (**<u>bold and underlined</u>**).

Rarefaction also slightly increased agreement between ASV and OTU-based beta diversity metrics, for both OTU thresholds (Table 2). Rarefaction had only a small effect on Bray-Curtis and Jaccard indices, with a correlation >0.9 between rarefied and unrarefied data and no effect on W-Unifrac (*r* = 1; S4 and S5 Figs in S1 File). Rarefaction did, however, have a greater influence on U-Unifrac values computed on OTU datasets (r = 0.82–0.87, S5 Fig in S1 File).

Methodological choices had different effects in detecting differences between communities. Random Forests tests on PERMANOVAs based on presence/absence indices (Jaccard and U-Unifrac) revealed that the choice of OTU vs. ASVs had a lower contribution in PERMANOVA's outcome compared to the other methodological choices (rarefaction level, beta diversity index) (S6 Fig in S1 File). Patterns based on abundance-weighted beta diversity indices (W-Unifrac and Bray-Curtis) had more contrasts, as the choice of ASV vs. OTUs had the lowest influence in mussel microbiome and sediment, but had the greatest contribution in seston communities (S6 Fig in S1 File).

For taxonomic beta diversity (Jaccard and Bray-Curtis indices), 97%-OTUs tended to detect higher differences across community types (Fig 4, $R^2$ = 0.23±0.06 in 97%-OTUs, 0.17±0.05 in 99%-OTUs, and 0.18±0.05 in ASVs). In contrast, for phylogenetic beta diversity (U- and W-Unifrac indices), ASVs-based PERMANOVAs showed a slightly higher distinction between sample types ($R^2$ = 0.31±0.11 in ASVs, 0.29±0.13 in 97%-OTUs, and 0.30±0.12 in 99%-OTUs) (Fig 4). ASVs-based metrics generally detected higher river and site effects than 97%- and 99%-OTUs ($R^2$ = 0.41±0.23 in ASVs, 0.38±0.21 in 97%-OTUs, and 0.39±0.21 in 99%-OTUs Fig 4).

Overall, PERMANOVAs outcomes were more variable across methodologies in seston than in the other sample types, while they were more homogeneous in mussel microbiome and sediment (Fig 4 and S7 Fig in S1 File). W-Unifrac raised the most consistent results across ASVs

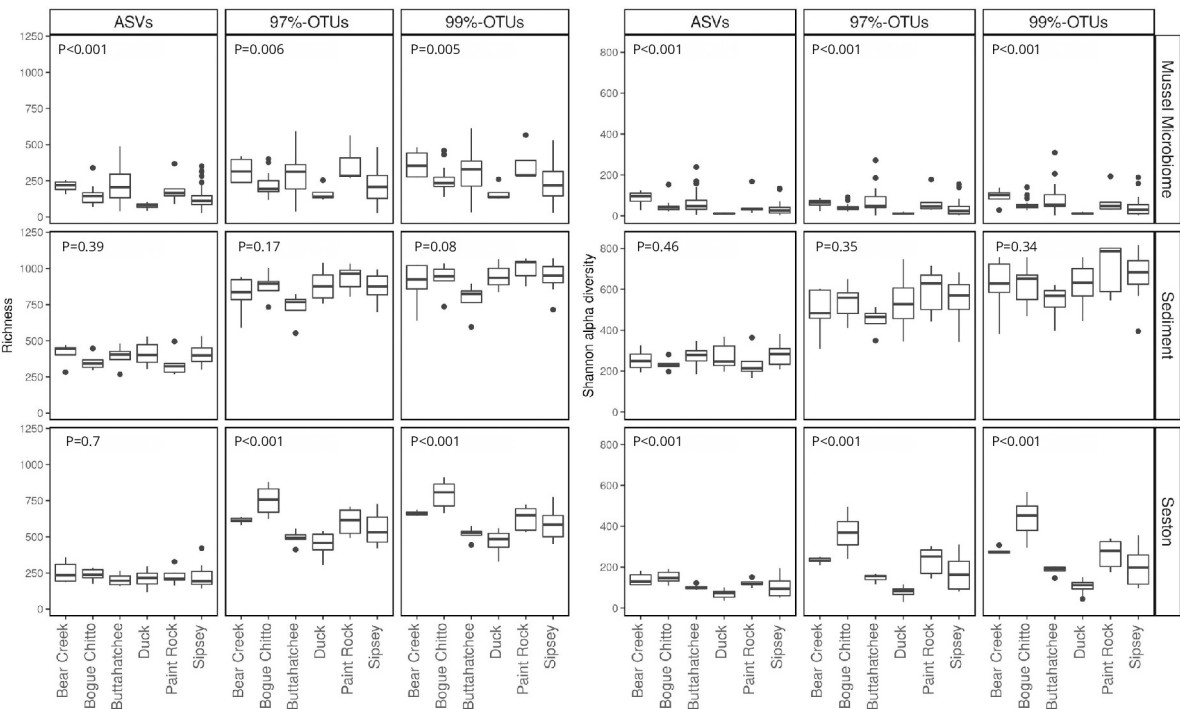

**Fig 3. Effect of sequence processing methodology on ecological patterns of microbial alpha diversity.** Distribution of alpha diversity values on each river is represented by a boxplot. Kruskal-Wallis tests assessed significant differences between rivers with p-values indicated on each panel. Types of communities are represented using horizontal panels. Patterns within ASVs, 97%-OTUs, and 99%-OTUs are compared using the three vertical panels on each plot. The richness index (left plot) and Shannon alpha diversity (right plot) are represented for each community type and sequence processing methodology.

vs. OTUs methodologies for detecting distinct community types (PERMANOVA's $R^2$ = 0.42 ±0.0), rivers ($R^2$ = 0.32±0.14), and sites ($R^2$ = 0.59±0.24) (S5f Fig in S1 File). The second most consistent index was Bray-Curtis ($R^2$ = 0.24±0.03, 0.34±0.14, and 0.57±0.21, respectively). Overall, the rarefaction level minimally influenced the PERMANOVA results (S7 Fig in S1 File).

## Composition

While Gammaproteobacteria (12.0±9.0%), Cyanobacteria (11.1±12.4%), Alphaproteobacteria (10.0±8.0%), and Planctomycetes (15.7±15.3%) were the main classes identified in all datasets and rarefaction levels used (Fig 5), several major classes of Bacteria had their relative abundance computed on OTUs poorly correlated with abundance computed on ASVs. These classes included Bacilli (Spearman's r = 0.34±0.21), Clostridia (r = 0.66±0.28), Bacteroidia (0.75 ±0.13), and Actinobacteria (0.81±0.17). Other major classes presented a higher reproducibility across methods (0.83±0.09). The class Verrucomicrobiae, which was dominant in OTU datasets (8.7±14% of relative abundance), was detected at a much lower level within ASVs, where it wasn't among the most common classes (1.7±0.6%) (Fig 5). Conversely, the ASV pipeline characterized Mollicutes within the mussel gut microbiome (13.4±13%), which remained completely undetected in the OTU datasets (Fig 5).

Differences across methods were more pronounced at the genus level, with only ~50% of the most common 30 genera shared between ASV and OTU-based datasets (excluding genera belonging to unclassified families), and correlations of the relative abundance of such shared genera generally below 0.9 (Spearman's correlation, Fig 5; S3 Table in S2 File). The 30 most

**Table 2. Correlation between beta-diversity metrics between OTU- and ASV-based datasets across every rarefaction level tested.**

| Beta-diversity index | Rarefaction level | 97%-OTUs vs. ASVs | 99%-OTUs vs. ASV |
|---|---|---|---|
| **Jaccard** | No rarefaction | 0.92 / 0.93 | 0.96 / 0.96 |
| | 3,000 | 0.93 / 0.93 | 0.96 / 0.96 |
| | 2,000 | 0.94 / 0.93 | 0.96 / 0.96 |
| | 1,000 | 0.94 / 0.93 | 0.96 / 0.95 |
| **Bray-Curtis** | No rarefaction | 0.92 / 0.93 | 0.96 / 0.97 |
| | 3,000 | 0.93 / 0.94 | 0.96 / 0.96 |
| | 2,000 | 0.94 / 0.94 | 0.97 / 0.96 |
| | 1,000 | 0.94 / 0.94 | 0.97 / 0.96 |
| **U-Unifrac** | No rarefaction | **0.80 / 0.80** | **0.82 / 0.83** |
| | 3,000 | **0.83 / 0.84** | **0.89 / 0.89** |
| | 2,000 | **0.84 / 0.83** | 0.90 / 0.90 |
| | 1,000 | **0.86 / 0.85** | 0.91 / 0.91 |
| **W-Unifrac** | No rarefaction | 0.93 / 0.94 | 0.95 / 0.96 |
| | 3,000 | 0.94 / 0.96 | 0.95 / 0.96 |
| | 2,000 | 0.94 / 0.95 | 0.95 / 0.96 |
| | 1,000 | 0.92 / 0.94 | 0.95 / 0.96 |

Correlations were assessed using Mantel tests, which correlation factors are indicated as follows Spearman's *r* / Pearson's *r* (all P values were < 0.001). Correlations <0.9 are highlighted in bold, showing poorer agreement between OTUs and ASVs. Correlations exhibiting poorer agreement between OTUs and ASVs are noted, *i.e. r*<0.9 (**bold**).

detected genera unique to ASV datasets included *Methyloglobus*, *Escherichia/Shigella*, and Clostridium sensus stricto (S3 Table in S2 File). Compared to the differences between ASV and OTU methodologies, rarefaction and OTU identity threshold had only a minor impact on overall class- or genus-level classification, with, for instance, a correlation of >0.9 in abundance of the major classes and genera between unrarefied and 2,000 sequence datasets, and between 99%- and 97% OTUs. As previously observed, with alpha and beta diversity indices, rarefaction had a lower impact on ASV- than OTU-based datasets, with all most detected genera shared before and after rarefaction (Fig 5).

## Discussion

Using a dataset that contained three types of microbial communities (mussel gut, sediment, and seston), we tested whether alpha and beta diversity metrics and community composition were consistent across sequences clustering methods, in addition to other methodological choices such as rarefaction and index. We detected significant influences of ASVs vs. OTUs on richness estimates, both on their values and the ranking of samples, as reported in previous studies [15, 19, 23]. Further, these differences resulted in discrepancies in the detection of biological signals. These discrepancies were accentuated by sampling bias corrected richness estimators like iChao1 and ACE, which, of all those tested, were the least correlated across the ASV and OTU datasets (Table 1). This discrepancy in diversity estimates was driven by the lower evenness of relative abundances and a much higher number of rare units with OTUs than ASVs (Fig 1). Hence, the large number of OTUs with low abundance likely induced greater variation in such estimators between OTU and ASV-based datasets, as such estimators use low-abundance units to infer the "real" richness within communities [54, 55]. As a consequence, the impact of ASVs vs. OTUs on the value of all richness measures was greater for

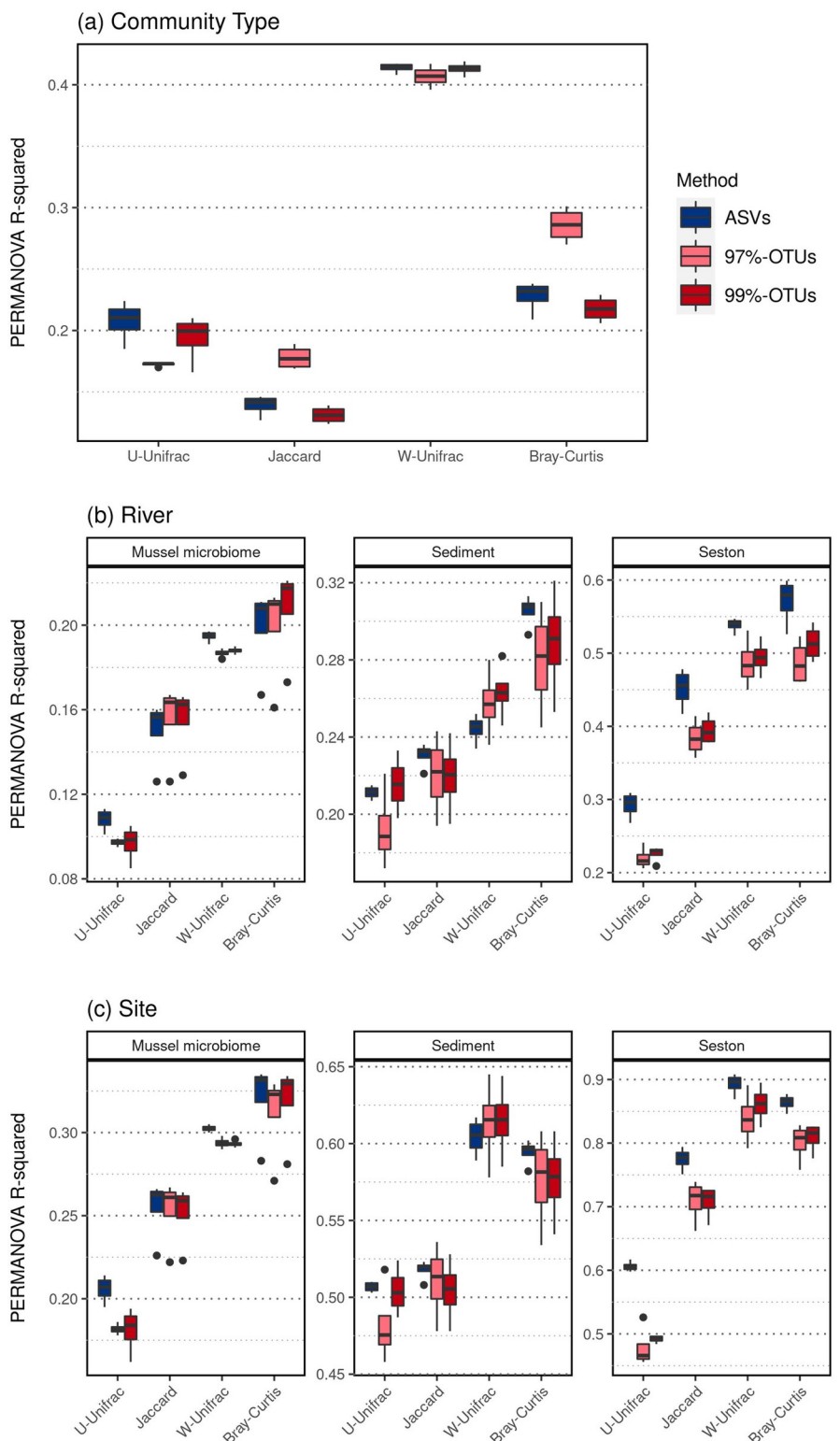

**Fig 4. Effect of sequence processing on the ecological signal based on beta-diversity estimates in microbial communities.** The intensity of the effect of (a) community type, (b) sampling river, and (c) collection site on community structure was assessed using separated PERMANOVAs for each factor, and each combination of sequence processing method x rarefaction level x index of dissimilarity. All rarefaction levels are aggregated on this figure. Differences across rarefaction levels are reported in S7 Fig in S1 File.

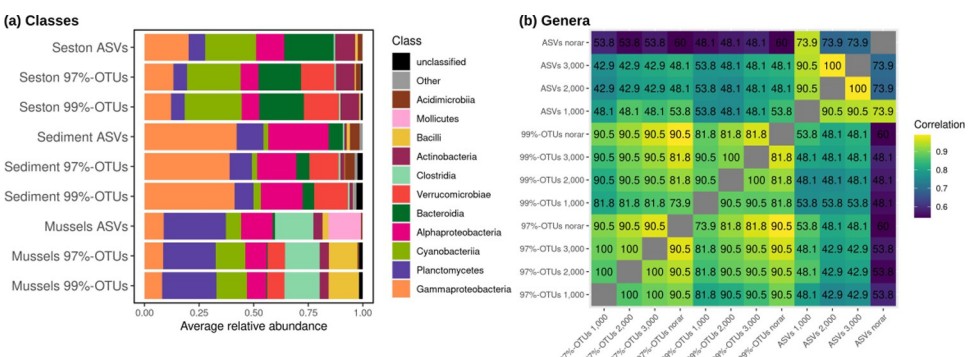

**Fig 5. Composition of mussel gut microbiome, sediment, and seston communities found using OTU-based and ASV-based methodologies.** (a) The relative abundance of the 11 most common bacterial classes in datasets rarefied to 2,000 sequences/sample. (b) Heatmap representing agreement of the 20 most detected bacterial genera across methodologies and rarefaction levels. Numbers represent the percentage of those 20 genera in common across the compared treatments ('shared'); color intensity represents overall Spearman's correlation coefficient of the shared genera between treatments.

environmental samples than for host-associated (mussel) microbiome (Fig 2 & S3 Fig in S1 File), due to the typical high microbial richness in environmental samples, especially sediment [56], contrary to gut mussel microbiomes which are dominated by fewer operational units [30, 57]. These findings contrast with Glassman and Martiny [14], who did not find a major effect of ASV vs. OTU-based pipelines on richness for bacterial communities sampled on leaf litter, which may be due to the lower richness in leaf litter environments. Purahong et al. [58] found a maximum richness in leaf litter of ~500 OTUs in similar leaf litter communities, while richness in our sediment samples averaged 2485±734 based on 97%-OTUs. In this context, the lower intrinsic richness of mussels' microbiomes may be an asset, as it would facilitate a correct estimation of richness measures and its comparison across studies based on distinct methodologies. However, a correlation of 0.70 in richness measured across pipelines highlights an important discrepancy between estimates for many samples as, for instance, 48% of these samples presented a variation >70% in their richness estimates between ASVs and OTUs before rarefaction.

Interestingly, in our study, a simple rarefaction of the OTU community tables increased the correlation between ASV- and OTU-based richness estimates, and decreased OTU-based richness to a level comparable to the estimate from ASVs (Table 1, Fig 2, S3 Fig in S1 File). While the use of rarefaction is considered a poor way of normalizing sequence data counts that can sometimes lead to false conclusions [25], it is also more likely to remove the rare operational units that could originate from sequencing errors, and may therefore be useful in richness estimation in OTU datasets [59]. In contrast, the impacts of ASV or OTU selection and rarefaction were much lower on Shannon entropy (with a correlation >0.9 at every rarefaction level studied, Table 1), and were comparable to previous comparisons on natural and mock communities [23, 60]. Indeed, the Shannon index accounts less for the rarest operational units than species richness metrics and thus is more robust to the difference in the number of rare units obtained with ASVs vs. OTUs clustering.

As with previous studies [14, 22, 23], we reported more consistency between OTU and ASV-based beta diversity metrics (Table 2). Beta diversity estimates based on 99%-OTUs were slightly more correlated to those based on 97%-OTUs, which is expected given the higher resolution of 99%- compared to 97%-OTUs, making them more comparable to ASVs [60]. Such correlation was also variable across beta diversity indices and was lower in the case of U-Unifrac, while it was higher in all other indices. This is consistent with previous observations from

Nearing et al. [15], who computed Bray-Curtis, U-Unifrac, and W-Unifrac dissimilarities on host-associated and soil datasets using several ASV-based pipelines and one OTU based pipeline. They also observed that U-Unifrac correlated poorly across pipelines, while the abundance-weighted indices were highly consistent.

We further compared the ecological signal raised by PERMANOVAs, testing the community type, river, and site's effect on ASV- and OTU-based datasets (Fig 4). Based on previous observations, distinction across sample types was expected, as these communities are extracted from very distinct environments [56, 61, 62]. Differences across rivers and sites were also expected in seston and sediment, as is typically the case in similar environmental communities [62]. Moderate differences across sites in the mussel microbiome were also previously observed by our group based on 97%-OTUs [30, 57].

According to the higher correlation between beta diversity estimates from ASVs and 99%-OTU datasets, PERMANOVA outcomes were also more similar between these two datasets, 99%-OTUs detecting higher river and site effects in seston and lower detection of the distinction between sample types than 97%-OTUs, consistent with patterns based on ASVs (Fig 4). The only exception to an overall high correlation between OTU- and ASV-based datasets was the U-Unifrac index, which showed a poor agreement between OTUs and ASV-based ecological signal (Fig 4, S7 Fig in S1 File). The choice of beta diversity index is crucial before a PERMANOVA, as while abundance weighted indices (taxonomic Bray-Curtis or phylogenetic W-Unifrac) are more sensitive to a change in dominant members of the community across contrasted conditions, indices based on presence-absence (taxonomic Jaccard or phylogenetic U-Unifrac) are more sensitive to the rare members of the community, and are therefore generally more useful in detecting subtle changes across near-identical communities [63]. Surprisingly in our study, the other presence-absence index tested, Jaccard, exhibited a higher correlation across ASVs vs. OTU methodologies (Fig 4, S7 Fig in S1 File), highlighting that in our dataset, differences across ASVs and OTUs are due to the rarest units, which are phylogenetically distinct from the more dominant ones. The Jaccard index ignores such phylogenetic information, allowing a more consistent pattern between OTUs and ASVs using this index.

Although clustering programs producing OTUs cluster more dissimilar sequences into the same unit than ASVs, previous authors suggested they could also generate a higher number of spurious, low-abundance OTUs that would inflate richness measures [15, 19]. Similarly, in a shrimp microbiome study, richness measurements were systematically higher in OTUs datasets compared to ASVs [60]. In our study, OTU-based datasets did generate up to 9 and 12 times (respectively for 97%- and 99%-OTUs) more OTUs than ASVs on the same samples before rarefaction (Fig 1, S1 Fig in S1 File). With the present data, we cannot conclude whether the high number of low-abundance, phylogenetically distinct OTUs are indeed erroneous. However, they certainly impacted the functioning of the U-Unifrac index, which detected lower ecological signals overall.

Prodan et al. [19] reported that applying a cutoff on sequence abundance within Mothur before constructing the OTUs was necessary for reducing the erroneous inflation of observed richness. The removal of singletons (OTUs represented by only one sequence) can drastically reduce the variation across sample replicates within the same sequencing run [64], underlining the sensitivity of OTU-based methods to sequencing errors. Including their removal may be important for OTU-based analyses to pre-filter potentially spurious, low abundance sequences that could induce the construction of numerous erroneous OTUs. However, in our study, rarefaction, while improving the agreement of alpha diversity estimates between OTUs and ASVs by removing low-abundance OTUs (Table 1), only marginally impacted the signal based on beta diversity, and showed no improvement of the ecological signal based on U-Unifrac in any dataset (Table 2, S7 Fig in S1 File). This suggests that additional filtering of sequences may be

needed for performing U-Unifrac on OTU datasets, or at least that patterns should be compared to those obtained from other beta diversity indices to identify potential bias due to rare units.

In addition to the influence of the analysis method on patterns in beta diversity, there were significant differences between ASVs and OTUs in terms of community composition, both at the genus and class level, including in the most abundant taxa (Fig 5, S3 Fig in S1 File). Similar discrepancies were highlighted by Straub et al. [23], who found that the relative abundance of several dominant taxa was poorly reproduced across both methodologies. We found that the OTU identity threshold and rarefaction level had only a marginal effect on the identity and the relative abundance of major classes and genera. Regarding the differences across methodologies, ASVs notably detected Mollicutes as preponderant in the mussel gut microbiome, which were absent in the OTU datasets. In contrast, OTUs detected much higher amounts of Verrucomicrobiae in the seston and sediment samples than were detected by the ASV approach. These differences are surprising given that both OTU and ASV approaches classified sequences to the same database. In this specific case, while the cultivation of Mollicutes is difficult [65], they have been successfully isolated from the digestive gland of various hosts, including aquatic invertebrates [66], making their detection within the gut of the mussels studied here plausible. On the other hand, Verrucomicrobia are also found in soil and water environments, and their presence is not surprising. Such discrepancies across dominant taxa are important to consider when comparing OTU- and ASV-based studies. While a previous study could increase the agreement between the two methodologies by applying a percentage cutoff to the operational units in the lowest abundance [60], in our case rarefaction was not sufficient to remove the discrepancies we found across both datasets.

## Conclusions

Using a dataset comprising environmental and host-associated samples, we revealed important differences in results obtained from an ASV-based denoising approach (DADA2) and an OTU-based clustering method (Mothur). We showed that the difference in tests' outcome and the ecological conclusion is predominantly due to rare operational units, which are more prevalent in OTU-based clustering approaches and therefore have more impact on indices computed only on presence/absence of units. However, there are also major differences in the occurrence and relative abundance of major bacterial taxa identified by both approaches. Other methodological choices (rarefaction, OTU identity threshold) had generally lower effects, although rarefaction and a clustering of 99%-identity OTUs tend to reconcile the differences between OTU- and ASV-based approaches.

## Supporting information

**S1 File.**
(DOCX)

**S2 File.**
(DOCX)

## Acknowledgments

Carla Atkinson, Garrett Hopper, Jamie Bucholz, Irene Sanchez Gonzalez, and Megan Kubala at the University of Alabama coordinated and assisted with the field collection of samples. Lauren Lawson at the University of Mississippi assisted with DNA extractions of mussel samples.

## Author Contributions

**Conceptualization:** Marlène Chiarello, Colin R. Jackson.

**Data curation:** Marlène Chiarello.

**Formal analysis:** Marlène Chiarello.

**Funding acquisition:** Colin R. Jackson.

**Investigation:** Marlène Chiarello, Mark McCauley.

**Methodology:** Marlène Chiarello, Sébastien Villéger.

**Project administration:** Colin R. Jackson.

**Supervision:** Colin R. Jackson.

**Validation:** Sébastien Villéger.

**Visualization:** Marlène Chiarello.

**Writing – original draft:** Marlène Chiarello.

**Writing – review & editing:** Marlène Chiarello, Mark McCauley, Sébastien Villéger, Colin R. Jackson.

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
