## [Decision Letter · Decision Letter 0]

29 Nov 2021

PONE-D-21-31893Ranking the biases: the choice of OTUs vs. ASVs in 16S rRNA amplicon data analysis has stronger effects on diversity measures than rarefaction and OTU identity thresholdPLOS ONE

Dear Dr. Chiarello,

Thank you for submitting your manuscript to PLOS ONE. After careful consideration, we feel that it has merit but does not fully meet PLOS ONE’s publication criteria as it currently stands. Therefore, we invite you to submit a revised version of the manuscript that addresses the points raised during the review process.

 The reviewer suggests a few, mostly editorial, modifications that should not be too much of a problem. Then the reviewer also suggests the use of a mock community, but does not think it's absolutely necessary. I agree, thus I leave the decision to add results using a mock community to you.

We look forward to receiving your revised manuscript.

Kind regards,

Gabriel Moreno-Hagelsieb

Academic Editor

PLOS ONE

Journal Requirements:

"Funding was provided from National Science Foundation award DEB 1831531 to C.R.J."

"Funding was provided from National Science Foundation (https://www.nsf.gov/) award DEB 1831531 to C.R.J. The funders had no role in study design, data collection and analysis, decision to publish, or preparation of the manuscript."

Reviewers' comments:

Reviewer's Responses to Questions

**Comments to the Author**

1. Is the manuscript technically sound, and do the data support the conclusions?

Reviewer #1: Partly

2. Has the statistical analysis been performed appropriately and rigorously? 

Reviewer #1: Yes

3. Have the authors made all data underlying the findings in their manuscript fully available?

Reviewer #1: Yes

4. Is the manuscript presented in an intelligible fashion and written in standard English?

Reviewer #1: Yes

5. Review Comments to the Author

Reviewer #1: The results provide some helpful considerations for the application and interpretation in 16S amplicon data studies.

The manuscript is presented with a clear structure and language.

The sequencing data underlying the findings will be available upon publication.

The applied statistical analyzes allow answering the problem posed.

1. Is the manuscript technically sound, and do the data support the conclusions?

Here are my comments:

Line 228 "groups across the different analysis treatments was made using Spearman’s signed rank tests."

I am not sure if you mean Spearman rank correlation test or Wilkoxon signed rank test.

Line 486. The discussion in this paragraph would be clearer if you cite the table/figures showing the corresponding information (correlation and ecological signal).

Line 509 "samples before rarefaction. In the absence of a mock community, we cannot conclude whether"

What is the reason you did not include a mock community? Although I don't consider this absolutely necessary for the traced objectives, even an in silico mock community would help to gain more insight into the subject.

Line 168 "... Forward and reverse reads with >2 or >5 estimated errors, respectively,"

Line 517 "underlining the sensitivity of OTU-based methods to sequencing errors "

You don’t provide information about the quality statistics from your sequencing data, however this is a central piece of information for both ASV performance and OTU. This information might also be valuable for the discussion.

Line 390 and 527. You found differences across both methodologies in some dominant taxa. Did you try to find the reasons behind such differences in your sequence dataset?

6. PLOS authors have the option to publish the peer review history of their article (what does this mean?). If published, this will include your full peer review and any attached files.

Reviewer #1: No

---

## [Author Response · Author response to Decision Letter 0]

24 Jan 2022

Response to reviewer’s comments:

Reviewer #1: The results provide some helpful considerations for the application and interpretation in 16S amplicon data studies.

The manuscript is presented with a clear structure and language.

The sequencing data underlying the findings will be available upon publication.

The applied statistical analyzes allow answering the problem posed.

Thank you for your time and such positive comments. We have answered these points that you raised below and highlighted the changes in the revised manuscript using the track change option.

1. Is the manuscript technically sound, and do the data support the conclusions?

Here are my comments:

Line 228 "groups across the different analysis treatments was made using Spearman’s signed rank tests."

I am not sure if you mean Spearman rank correlation test or Wilkoxon signed rank test.

>We used both tests. The Wilcoxon signed-rank test for repeated measurements tested whether there was a significantly different alpha diversity value on the same samples after distinct sequence processing methods (e.g. Mothur or DADA2). Spearman’s and Pearson’s correlations were then computed to measure the degree of correlation of such measures across methods. We modified the text to clarify this point.

Line 486. The discussion in this paragraph would be clearer if you cite the table/figures showing the corresponding information (correlation and ecological signal).

>We agree and have added this information.

Line 509 "samples before rarefaction. In the absence of a mock community, we cannot conclude whether"

What is the reason you did not include a mock community? Although I don't consider this absolutely necessary for the traced objectives, even an in silico mock community would help to gain more insight into the subject.

>The idea of this study originated after we confronted the results of our newer assessment of mussel microbiome composition after switching to ASV-based methods, with our previous OTU-based studies. As we observed major differences, we decided to make a more systematic comparison based on our newest set of samples, which resulted in the present manuscript. As this study wasn’t initially planned, we did not include a mock community in our sequencing run, which would have helped identifying potential spurious OTUs. However, given the much higher number of OTUs compared to ASVs, and lower evenness of OTU-based communities, we think that our hypothesis of richness overestimation by Mothur is reasonable, especially as it agrees with previous results based on mock communities (e.g. Prodan et al, 2020, and others cited in MS). We agree that a simulated sequencing run (e.g. using the tool from Richter et al, 2008) that would be processed by both ASV- and OTU-based methods, could help in understanding the finer difference between them, but it would deserve an entirely new study, as it would require (i) a long computation time, and (ii) a tricky calibration as the bacteria in our communities of study generally lack representatives in genome databases.

Therefore, we removed the mention of a mock community within the discussion and simplified the text L520.

Line 168 "... Forward and reverse reads with >2 or >5 estimated errors, respectively,"

>Corrected

Line 517 "underlining the sensitivity of OTU-based methods to sequencing errors "

You don’t provide information about the quality statistics from your sequencing data, however this is a central piece of information for both ASV performance and OTU. This information might also be valuable for the discussion.

>We added a table with such statistics in Supplementary Table S3. All our samples passed the fastqc standard for sequence quality scores. Globally the statistics were comparable to our previous assessments from environmental samples.

Line 390 and 527. You found differences across both methodologies in some dominant taxa. Did you try to find the reasons behind such differences in your sequence dataset?

>We blast-searched the representative sequences of Mollicutes and Verrucomicrobiae over the NCBI database. From the results, our identifications were correct. Our main hypothesis is that differences in sequence processing led to the removal of sequences that were distinct between Mothur and DADA2. It is however difficult to investigate this further without using appropriate mock communities.

>References:

>Richter, D. C., Ott, F., Auch, A. F., Schmid, R., & Huson, D. H. (2008). MetaSim—a sequencing simulator for genomics and metagenomics. PloS one, 3(10), e3373.

>Prodan A, Tremaroli V, Brolin H, Zwinderman AH, Nieuwdorp M, Levin E. Comparing Bioinformatic Pipelines for Microbial 16S rRNA Amplicon Sequencing. Seo J-S, editor. PLOS ONE. 2020 Jan 16;15(1):e0227434.

---

## [Editor Report · Decision Letter 1]

11 Feb 2022

Ranking the biases: the choice of OTUs vs. ASVs in 16S rRNA amplicon data analysis has stronger effects on diversity measures than rarefaction and OTU identity threshold

PONE-D-21-31893R1

Dear Dr. Chiarello,

We’re pleased to inform you that your manuscript has been judged scientifically suitable for publication and will be formally accepted for publication once it meets all outstanding technical requirements.

Kind regards,

Gabriel Moreno-Hagelsieb

Academic Editor

PLOS ONE
---

## [Editor Report · Acceptance letter]

15 Feb 2022

PONE-D-21-31893R1 

Ranking the biases: the choice of OTUs vs. ASVs in 16S rRNA amplicon data analysis has stronger effects on diversity measures than rarefaction and OTU identity threshold 

Dear Dr. Chiarello:

I'm pleased to inform you that your manuscript has been deemed suitable for publication in PLOS ONE. Congratulations! Your manuscript is now with our production department. 

Kind regards, 

on behalf of

Prof. Gabriel Moreno-Hagelsieb 

Academic Editor

PLOS ONE